# Unsupervised Machine Translation Using Monolingual Corpora Only

**Guillaume Lample** † ‡ , **Alexis Conneau** † , **Ludovic Denoyer** ‡ , **Marc'Aurelio Ranzato** †
† Facebook AI Research,
‡ Sorbonne Universités, UPMC Univ Paris 06, LIP6 UMR 7606, CNRS
{gl,aconneau,ranzato}@fb.com,ludovic.denoyer@lip6.fr

## ABSTRACT

Machine translation has recently achieved impressive performance thanks to recent advances in deep learning and the availability of large-scale parallel corpora. There have been numerous attempts to extend these successes to low-resource language pairs, yet requiring tens of thousands of parallel sentences. In this work, we take this research direction to the extreme and investigate whether it is possible to learn to translate even *without any* parallel data. We propose a model that takes sentences from monolingual corpora in two different languages and maps them into the same latent space. By learning to reconstruct in both languages from this shared feature space, the model effectively learns to translate without using any labeled data. We demonstrate our model on two widely used datasets and two language pairs, reporting BLEU scores of 32.8 and 15.1 on the Multi30k and WMT English-French datasets, without using even a single parallel sentence at training time.

## 1 INTRODUCTION

Thanks to recent advances in deep learning (Sutskever et al., 2014; Bahdanau et al., 2015) and the availability of large-scale parallel corpora, machine translation has now reached impressive performance on several language pairs (Wu et al., 2016). However, these models work very well only when provided with massive amounts of parallel data, in the order of millions of parallel sentences. Unfortunately, parallel corpora are costly to build as they require specialized expertise, and are often nonexistent for low-resource languages. Conversely, monolingual data is much easier to find, and many languages with limited parallel data still possess significant amounts of monolingual data.

There have been several attempts at leveraging monolingual data to improve the quality of machine translation systems in a semi-supervised setting (Munteanu et al., 2004; Irvine, 2013; Irvine & Callison-Burch, 2015; Zheng et al., 2017). Most notably, Sennrich et al. (2015a) proposed a very effective data-augmentation scheme, dubbed "back-translation", whereby an auxiliary translation system from the target language to the source language is first trained on the available parallel data, and then used to produce translations from a large monolingual corpus on the target side. The pairs composed of these translations with their corresponding ground truth targets are then used as additional training data for the original translation system.

Another way to leverage monolingual data on the target side is to augment the decoder with a language model (Gulcehre et al., 2015). And finally, Cheng et al. (2016); He et al. (2016) have proposed to add an auxiliary auto-encoding task on monolingual data, which ensures that a translated sentence can be translated back to the original one. All these works still rely on several tens of thousands parallel sentences, however.

Previous work on zero-resource machine translation has also relied on labeled information, not from the language pair of interest but from other related language pairs (Firat et al., 2016; Johnson et al., 2016; Chen et al., 2017) or from other modalities (Nakayama & Nishida, 2017; Lee et al., 2017). The only exception is the work by Ravi & Knight (2011); Pourdamghani & Knight (2017), where the machine translation problem is reduced to a deciphering problem. Unfortunately, their method is limited to rather short sentences and it has only been demonstrated on a very simplistic setting comprising of the most frequent short sentences, or very closely related languages.

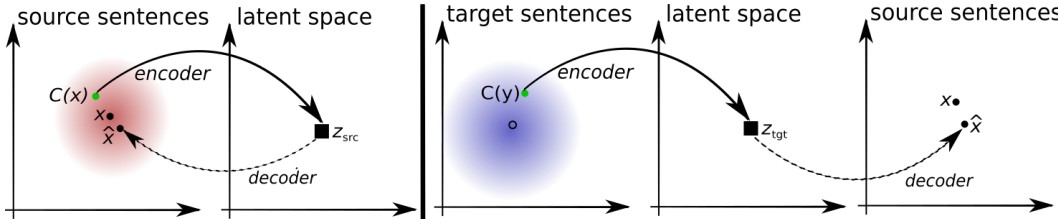

Figure 1: Toy illustration of the principles guiding the design of our objective function. Left (auto-encoding): the model is trained to reconstruct a sentence from a noisy version of it. $x$ is the target, $C(x)$ is the noisy input, $\hat{x}$ is the reconstruction. Right (translation): the model is trained to translate a sentence in the other domain. The input is a noisy translation (in this case, from source-to-target) produced by the model itself, $M$, at the previous iteration $(t)$, $y = M^{(t)}(x)$. The model is symmetric, and we repeat the same process in the other language. See text for more details.

In this paper, we investigate whether it is possible to train a general machine translation system without any form of supervision whatsoever. The only assumption we make is that there exists a monolingual corpus on each language. This set up is interesting for a twofold reason. First, this is applicable whenever we encounter a new language pair for which we have no annotation. Second, it provides a strong lower bound performance on what any good semi-supervised approach is expected to yield.

The key idea is to build a common latent space between the two languages (or domains) and to learn to translate by reconstructing in both domains according to two principles: (i) the model has to be able to reconstruct a sentence in a given language from a noisy version of it, as in standard denoising auto-encoders (Vincent et al., 2008). (ii) The model also learns to reconstruct any source sentence given a noisy translation of the same sentence in the target domain, and vice versa. For (ii), the translated sentence is obtained by using a back-translation procedure (Sennrich et al., 2015a), i.e. by using the learned model to translate the source sentence to the target domain. In addition to these reconstruction objectives, we constrain the source and target sentence latent representations to have the same distribution using an adversarial regularization term, whereby the model tries to fool a discriminator which is simultaneously trained to identify the language of a given latent sentence representation (Ganin et al., 2016). This procedure is then iteratively repeated, giving rise to translation models of increasing quality. To keep our approach fully unsupervised, we initialize our algorithm by using a naïve unsupervised translation model based on a word by word translation of sentences with a bilingual lexicon derived from the same monolingual data (Conneau et al., 2017). As a result, and by only using monolingual data, we can encode sentences of both languages into the same feature space, and from there, we can also decode/translate in any of these languages; see Figure 1 for an illustration.

While not being able to compete with supervised approaches using lots of parallel resources, we show in section 4 that our model is able to achieve remarkable performance. For instance, on the WMT dataset we can achieve the same translation quality of a similar machine translation system trained with full supervision on 100,000 sentence pairs. On the Multi30K-Task1 dataset we achieve a BLEU above 22 on all the language pairs, with up to 32.76 on English-French.

Next, in section 2, we describe the model and the training algorithm. We then present experimental results in section 4. Finally, we further discuss related work in section 5 and summarize our findings in section 6.

## 2 UNSUPERVISED NEURAL MACHINE TRANSLATION

In this section, we first describe the architecture of the translation system, and then we explain how we train it.

## 2.1 Neural Machine Translation Model

The translation model we propose is composed of an encoder and a decoder, respectively responsible for encoding source and target sentences to a latent space, and to decode from that latent space to the source or the target domain. We use a single encoder and a single decoder for both domains (Johnson et al., 2016). The only difference when applying these modules to different languages is the choice of lookup tables.

Let us denote by $\mathcal{W}_S$ the set of words in the source domain associated with the (learned) words embeddings $\mathcal{Z}^S = (z_1^s, ...., z_{|\mathcal{W}_S|}^s)$, and by $\mathcal{W}_T$ the set of words in the target domain associated with the embeddings $\mathcal{Z}^T = (z_1^t, ...., z_{|\mathcal{W}_T|}^t)$, $\mathcal{Z}$ being the set of all the embeddings. Given an input sentence of $m$ words $\boldsymbol{x} = (x_1, x_2, ..., x_m)$ in a particular language $\ell$, $\ell \in \{src, tgt\}$, an encoder $e_{\theta_{enc}, \mathcal{Z}}(\boldsymbol{x}, \ell)$ computes a sequence of $m$ hidden states $\boldsymbol{z} = (z_1, z_2, ..., z_m)$ by using the corresponding word embeddings, i.e. $\mathcal{Z}_S$ if $\ell = src$ and $\mathcal{Z}_T$ if $\ell = tgt$; the other parameters $\theta_{enc}$ are instead shared between the source and target languages. For the sake of simplicity, the encoder will be denoted as $e(\boldsymbol{x}, \ell)$ in the following. These hidden states are vectors in $\mathbb{R}^n$, $n$ being the dimension of the latent space.

A decoder $d_{\theta_{dec}, \mathcal{Z}}(\boldsymbol{z}, \ell)$ takes as input $\boldsymbol{z}$ and a language $\ell$, and generates an output sequence $\boldsymbol{y} = (y_1, y_2, ..., y_k)$, where each word $y_i$ is in the corresponding vocabulary $\mathcal{W}^\ell$. This decoder makes use of the corresponding word embeddings, and it is otherwise parameterized by a vector $\theta_{dec}$ that does not depend on the output language. It will thus be denoted $d(\boldsymbol{z}, \ell)$ in the following. To generate an output word $y_i$, the decoder iteratively takes as input the previously generated word $y_{i-1}$ ($y_0$ being a start symbol which is language dependent), updates its internal state, and returns the word that has the highest probability of being the next one. The process is repeated until the decoder generates a stop symbol indicating the end of the sequence.

In this article, we use a sequence-to-sequence model with attention (Bahdanau et al., 2015), without input-feeding. The encoder is a bidirectional-LSTM which returns a sequence of hidden states $\boldsymbol{z} = (z_1, z_2, ..., z_m)$. At each step, the decoder, which is also an LSTM, takes as input the previous hidden state, the current word and a context vector given by a weighted sum over the encoder states. In all the experiments we consider, both encoder and decoder have 3 layers. The LSTM layers are shared between the source and target encoder, as well as between the source and target decoder. We also share the attention weights between the source and target decoder. The embedding and LSTM hidden state dimensions are all set to 300. Sentences are generated using greedy decoding.

## 2.2 Overview of the Method

We consider a dataset of sentences in the source domain, denoted by $\mathcal{D}_{src}$, and another dataset in the target domain, denoted by $\mathcal{D}_{tgt}$. These datasets do not correspond to each other, in general. We train the encoder and decoder by reconstructing a sentence in a particular domain, given a noisy version of the same sentence in the same or in the other domain.

At a high level, the model starts with an unsupervised naïve translation model obtained by making word-by-word translation of sentences using a parallel dictionary learned in an unsupervised way (Conneau et al., 2017). Then, at each iteration, the encoder and decoder are trained by minimizing an objective function that measures their ability to both reconstruct and translate from a noisy version of an input training sentence. This noisy input is obtained by dropping and swapping words in the case of the auto-encoding task, while it is the result of a translation with the model at the previous iteration in the case of the translation task. In order to promote alignment of the latent distribution of sentences in the source and the target domains, our approach also simultaneously learns a discriminator in an adversarial setting. The newly learned encoder/decoder are then used at the next iteration to generate new translations, until convergence of the algorithm. At test time and despite the lack of parallel data at training time, the encoder and decoder can be composed into a standard machine translation system.

## 2.3 Denoising auto-encoding

Training an autoencoder of sentences is a trivial task, if the sequence-to-sequence model is provided with an attention mechanism like in our work [1]. Without any constraint, the auto-encoder very quickly learns to merely copy every input word one by one. Such a model would also perfectly copy sequences of random words, suggesting that the model does not learn any useful structure in the data. To address this issue, we adopt the same strategy of Denoising Auto-encoders (DAE) (Vincent et al., 2008)), and add noise to the input sentences (see Figure 1-left), similarly to Hill et al. (2016). Considering a domain $\ell = src$ or $\ell = tgt$, and a stochastic noise model denoted by $C$ which operates on sentences, we define the following objective function:

$$\mathcal{L}_{auto}(\theta_{\text{enc}}, \theta_{\text{dec}}, \mathcal{Z}, \ell) = \mathbb{E}_{x \sim \mathcal{D}_\ell, \hat{x} \sim d(e(C(x), \ell), \ell)} \left[ \Delta(\hat{x}, x) \right] \tag{1}$$

where $\hat{x} \sim d(e(C(x), \ell), \ell)$ means that $\hat{x}$ is a reconstruction of the corrupted version of $x$, with $x$ sampled from the monolingual dataset $\mathcal{D}_\ell$. In this equation, $\Delta$ is a measure of discrepancy between the two sequences, the sum of token-level cross-entropy losses in our case.

**Noise model** $C(x)$ is a randomly sampled noisy version of sentence $x$. In particular, we add two different types of noise to the input sentence. First, we drop every word in the input sentence with a probability $p_{wd}$. Second, we *slightly* shuffle the input sentence. To do so, we apply a random permutation $\sigma$ to the input sentence, verifying the condition $\forall i \in \{1, n\}, |\sigma(i) - i| \le k$ where $n$ is the length of the input sentence, and $k$ is a tunable parameter.

To generate a random permutation verifying the above condition for a sentence of size $n$, we generate a random vector $q$ of size $n$, where $q_i = i + U(0, \alpha)$, and $U$ is a draw from the uniform distribution in the specified range. Then, we define $\sigma$ to be the permutation that sorts the array $q$. In particular, $\alpha < 1$ will return the identity, $\alpha = +\infty$ can return any permutation, and $\alpha = k + 1$ will return permutations $\sigma$ verifying $\forall i \in \{1, n\}, |\sigma(i) - i| \le k$. Although biased, this method generates permutations similar to the noise observed with word-by-word translation.

In our experiments, both the word dropout and the input shuffling strategies turned out to have a critical impact on the results, see also section 4.5, and using both strategies at the same time gave us the best performance. In practice, we found $p_{wd} = 0.1$ and $k = 3$ to be good parameters.

## 2.4 Cross Domain Training

The second objective of our approach is to constrain the model to be able to map an input sentence from a the source/target domain $\ell_1$ to the target/source domain $\ell_2$, which is what we are ultimately interested in at test time. The principle here is to sample a sentence $x \in \mathcal{D}_{\ell_1}$, and to generate a corrupted translation of this sentence in $\ell_2$. This corrupted version is generated by applying the current translation model denoted $M$ to $x$ such that $y = M(x)$. Then a corrupted version $C(y)$ is sampled (see Figure 1-right). The objective is thus to learn the encoder and the decoder such that they can reconstruct $x$ from $C(y)$. The cross-domain loss can be written as:

$$\mathcal{L}_{cd}(\theta_{\text{enc}}, \theta_{\text{dec}}, \mathcal{Z}, \ell_1, \ell_2) = \mathbb{E}_{x \sim \mathcal{D}_{\ell_1}, \hat{x} \sim d(e(C(M(x)), \ell_2), \ell_1)} \left[ \Delta(\hat{x}, x) \right] \tag{2}$$

where $\Delta$ is again the sum of token-level cross-entropy losses.

## 2.5 Adversarial training

Intuitively, the decoder of a neural machine translation system works well only when its input is produced by the encoder it was trained with, or at the very least, when that input comes from a distribution very close to the one induced by its encoder. Therefore, we would like our encoder to output features in the same space regardless of the actual language of the input sentence. If such condition is satisfied, our decoder may be able to decode in a certain language regardless of the language of the encoder input sentence.

Note however that the decoder could still produce a bad translation while yielding a valid sentence in the target domain, as constraining the encoder to map two languages in the same feature space does

---

[1]Even without attention, reconstruction can be surprisingly easy, depending on the length of the input sentence and the dimensionality of the embeddings, as suggested by concentration of measure and theory of sparse recovery (Donoho, 2006).

not imply a strict correspondence between sentences. Fortunately, the previously introduced loss for cross-domain training in equation 2 mitigates this concern. Also, recent work on bilingual lexical induction has shown that such a constraint is very effective at the word level, suggesting that it may also work at the sentence level, as long as the two latent representations exhibit strong structure in feature space.

In order to add such a constraint, we train a neural network, which we will refer to as the *discriminator*, to classify between the encoding of source sentences and the encoding of target sentences (Ganin et al., 2016). The discriminator operates on the output of the encoder, which is a sequence of latent vectors $(z_1, ..., z_m)$, with $z_i \in \mathbb{R}^n$, and produces a binary prediction about the language of the encoder input sentence: $p_D(l|z_1, ..., z_m) \propto \prod_{j=1}^{m} p_D(\ell|z_j)$, with $p_D : \mathbb{R}^n \rightarrow [0; 1]$, where 0 corresponds to the source domain, and 1 to the target domain.

The discriminator is trained to predict the language by minimizing the following cross-entropy loss: $\mathcal{L}_{\mathcal{D}}(\theta_D|\theta, \mathcal{Z}) = -\mathbb{E}_{(x_i, \ell_i)}[\log p_D(\ell_i|e(x_i, \ell_i))]$, where $(x_i, \ell_i)$ corresponds to sentence and language id pairs uniformly sampled from the two monolingual datasets, $\theta_D$ are the parameters of the discriminator, $\theta_{\text{enc}}$ are the parameters of the encoder, and $\mathcal{Z}$ are the encoder word embeddings.

The encoder is trained instead to fool the discriminator:

$$\mathcal{L}_{adv}(\theta_{\text{enc}}, \mathcal{Z}|\theta_D) = -\mathbb{E}_{(x_i, \ell_i)}[\log p_D(\ell_j|e(x_i, \ell_i))] \tag{3}$$

with $\ell_j = \ell_1$ if $\ell_i = \ell_2$, and vice versa.

**Final Objective function**    The final objective function at one iteration of our learning algorithm is thus:

$$\begin{aligned}
\mathcal{L}(\theta_{\text{enc}}, \theta_{\text{dec}}, \mathcal{Z}) =&\lambda_{auto}[\mathcal{L}_{auto}(\theta_{\text{enc}}, \theta_{\text{dec}}, \mathcal{Z}, src) + \mathcal{L}_{auto}(\theta_{\text{enc}}, \theta_{\text{dec}}, \mathcal{Z}, tgt)]+ \\
&\lambda_{cd}[\mathcal{L}_{cd}(\theta_{\text{enc}}, \theta_{\text{dec}}, \mathcal{Z}, src, tgt) + \mathcal{L}_{cd}(\theta_{\text{enc}}, \theta_{\text{dec}}, \mathcal{Z}, tgt, src)]+ \\
&\lambda_{adv}\mathcal{L}_{adv}(\theta_{\text{enc}}, \mathcal{Z}|\theta_D)
\end{aligned} \tag{4}$$

where $\lambda_{auto}$, $\lambda_{cd}$, and $\lambda_{adv}$ are hyper-parameters weighting the importance of the auto-encoding, cross-domain and adversarial loss. In parallel, the discriminator loss $\mathcal{L}_D$ is minimized to update the discriminator.

## 3    TRAINING

In this section we describe the overall training algorithm and the unsupervised criterion we used to select hyper-parameters.

### 3.1    ITERATIVE TRAINING

The final learning algorithm is described in Algorithm 1 and the general architecture of the model is shown in Figure 2. As explained previously, our model relies on an iterative algorithm which starts from an initial translation model $M^{(1)}$ (line 3). This is used to translate the available monolingual data, as needed by the cross-domain loss function of Equation 2. At each iteration, a new encoder and decoder are trained by minimizing the loss of Equation 4 – line 7 of the algorithm. Then, a new translation model $M^{(t+1)}$ is created by composing the resulting encoder and decoder, and the process repeats.

To jump start the process, $M^{(1)}$ simply makes a word-by-word translation of each sentence using a parallel dictionary learned using the unsupervised method proposed by Conneau et al. (2017), which only leverages monolingual data.

The intuition behind our algorithm is that as long as the initial translation model $M^{(1)}$ retains at least some information of the input sentence, the encoder will map such translation into a representation in feature space that also corresponds to a cleaner version of the input, since the encoder is trained to denoise. At the same time, the decoder is trained to predict noiseless outputs, conditioned on noisy features. Putting these two pieces together will produce less noisy translations, which will enable better back-translations at the next iteration, and so on so forth.

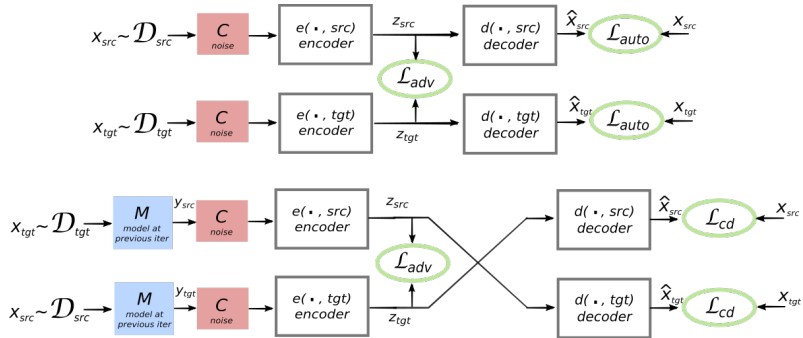

Figure 2: Illustration of the proposed architecture and training objectives. The architecture is a sequence to sequence model, with both encoder and decoder operating on two languages depending on an input language identifier that swaps lookup tables. Top (auto-encoding): the model learns to denoise sentences in each domain. Bottom (translation): like before, except that we encode from another language, using as input the translation produced by the model at the previous iteration (light blue box). The green ellipses indicate terms in the loss function.

---

**Algorithm 1** Unsupervised Training for Machine Translation

---

 1: **procedure** TRAINING($\mathcal{D}_{src}, \mathcal{D}_{tgt}, T$)
 2:    Infer bilingual dictionary using monolingual data (Conneau et al., 2017)
 3:    $M^{(1)} \leftarrow$ unsupervised word-by-word translation model using the inferred dictionary
 4:    **for** $t = 1, T$ **do**
 5:       using $M^{(t)}$, translate each monolingual dataset
 6:       // discriminator training & model training as in eq. 4
 7:       $\theta_{discr} \leftarrow \arg \min \mathcal{L}_D, \quad \theta_{enc}, \theta_{dec}, \mathcal{Z} \leftarrow \arg \min \mathcal{L}$
 8:       $M^{(t+1)} \leftarrow e^{(t)} \circ d^{(t)}$ // update MT model
 9:    **end for**
10:    return $M^{(T+1)}$
11: **end procedure**

---

## 3.2 Unsupervised Model Selection Criterion

In order to select hyper-parameters, we wish to have a criterion correlated with the translation quality. However, we do not have access to parallel sentences to judge how well our model translates, not even at validation time. Therefore, we propose the surrogate criterion which we show correlates well with BLEU (Papineni et al., 2002), the metric we care about at test time.

For all sentences $x$ in a domain $\ell_1$, we translate these sentences to the other domain $\ell_2$, and then translate the resulting sentences back to $\ell_1$. The quality of the model is then evaluated by computing the BLEU score over the original inputs and their reconstructions via this two-step translation process. The performance is then averaged over the two directions, and the selected model is the one with the highest average score.

Given an encoder $e$, a decoder $d$ and two non-parallel datasets $\mathcal{D}_{src}$ and $\mathcal{D}_{tgt}$, we denote $M_{src \to tgt}(x) = d(e(x, src), tgt)$ the translation model from *src* to *tgt*, and $M_{tgt \to src}$ the model in the opposite direction. Our model selection criterion $MS(e, d, \mathcal{D}_{src}, \mathcal{D}_{tgt})$ is:

$$
\begin{aligned}
MS(e, d, \mathcal{D}_{src}, \mathcal{D}_{tgt}) \quad = \quad & \frac{1}{2} \mathbb{E}_{x \sim \mathcal{D}_{src}} \left[ \mathrm{BLEU}(x, M_{src \to tgt} \circ M_{tgt \to src}(x)) \right] + \\
& \frac{1}{2} \mathbb{E}_{x \sim \mathcal{D}_{tgt}} \left[ \mathrm{BLEU}(x, M_{tgt \to src} \circ M_{src \to tgt}(x)) \right]
\end{aligned}
\tag{5}
$$

Figure 3 shows a typical example of the correlation between this measure and the final translation model performance (evaluated here using a parallel dataset).

The unsupervised model selection criterion is used both to a) determine when to stop training and b) to select the best hyper-parameter setting across different experiments. In the former case, the

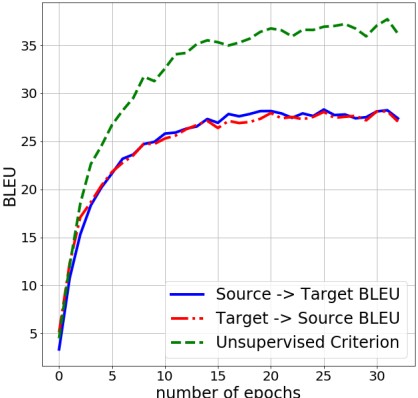

Figure 3: **Unsupervised model selection.** BLEU score of the source to target and target to source models on the Multi30k-Task1 English-French dataset as a function of the number of passes through the dataset at iteration $(t) = 1$ of the algorithm (training $M(2)$ given $M(1)$). BLEU correlates very well with the proposed model selection criterion, see Equation 5.

Spearman correlation coefficient between the proposed criterion and BLEU on the test set is 0.95 in average. In the latter case, the coefficient is in average 0.75, which is fine but not nearly as good. For instance, the BLEU score on the test set of models selected with the unsupervised criterion are sometimes up to 1 or 2 BLEU points below the score of models selected using a small validation set of 500 parallel sentences.

## 4 EXPERIMENTS

In this section, we first describe the datasets and the pre-processing we used, then we introduce the baselines we considered, and finally we report the extensive empirical validation proving the effectiveness of our method. We will release the code to the public once the revision process is over.

### 4.1 DATASETS

In our experiments, we consider the English-French and English-German language pairs, on three different datasets.

**WMT'14 English-French**    We use the full training set of 36 million pairs, we lower-case them and remove sentences longer than 50 words, as well as pairs with a source/target length ratio above 1.5, resulting in a parallel corpus of about 30 million sentences. Next, we build monolingual corpora by selecting the English sentences from 15 million random pairs, and selecting the French sentences from the complementary set. The former set constitutes our English monolingual dataset. The latter set is our French monolingual dataset. The lack of overlap between the two sets ensures that there is not exact correspondence between examples in the two datasets.

The validation set is comprised of 3,000 English and French sentences extracted from our monolingual training corpora described above. These sentences are not the translation of each other, and they will be used by our unsupervised model selection criterion, as explained in 3.2. Finally, we report results on the full *newstest2014* dataset.

**WMT'16 English-German**    We follow the same procedure as above to create monolingual training and validation corpora in English and German, which results in two monolingual training corpora of 1.8 million sentences each. We test our model on the *newstest2016* dataset.

**Multi30k-Task1**    The task 1 of the Multi30k dataset (Elliott et al., 2016) has 30,000 images, with annotations in English, French and German, that are translations of each other. We consider the English-French and English-German pairs. We disregard the images and only consider the parallel annotations, with the provided training, validation and test sets, composed of 29,000, 1,000 and 1,000 pairs of sentences respectively. For both pairs of languages and similarly to the WMT datasets

|  | MMT1 en-fr | MMT1 de-en | WMT en-fr | WMT de-en |
|---|---|---|---|---|
| Monolingual sentences | 14.5k | 14.5k | 15M | 1.8M |
| Vocabulary size | 10k / 11k | 19k / 10k | 67k / 78k | 80k / 46k |

Table 1: **Multi30k-Task1 and WMT datasets statistics.** To limit the vocabulary size in the WMT en-fr and WMT de-en datasets, we only considered words with more than 100 and 25 occurrences, respectively.

above, we split the training and validation sets into monolingual corpora, resulting in 14,500 monolingual source and target sentences in the training set, and 500 sentences in the validation set.

Table 1 summarizes the number of monolingual sentences in each dataset, along with the vocabulary size. To limit the vocabulary size on the WMT en-fr and WMT de-en datasets, we only considered words with more than 100 and 25 occurrences, respectively.

## 4.2 BASELINES

**Word-by-word translation (WBW)**   The first baseline is a system that performs word-by-word translations of the input sentences using the inferred bilingual dictionary (Conneau et al., 2017). This baseline provides surprisingly good results for related language pairs, like English-French, where the word order is similar, but performs rather poorly on more distant pairs like English-German, as can be seen in Table 2.

**Word reordering (WR)**   After translating word-by-word as in WBW, here we reorder words using an LSTM-based language model trained on the target side. Since we cannot exhaustively score every possible word permutation (some sentences have about 100 words), we consider all pairwise swaps of neighboring words, we select the best swap, and iterate ten times. We use this baseline only on the WMT dataset that has a large enough monolingual data to train a language model.

**Oracle Word Reordering (OWR)**   Using the reference, we produce the best possible generation using only the words given by WBW. The performance of this method is an *upper-bound* of what any model could do without replacing words.

**Supervised Learning**   We finally consider exactly the same model as ours, but trained with supervision, using the standard cross-entropy loss on the original parallel sentences.

## 4.3 UNSUPERVISED DICTIONARY LEARNING

To implement our baseline and also to initialize the embeddings $\mathcal{Z}$ of our model, we first train word embeddings on the source and target monolingual corpora using fastText (Bojanowski et al., 2017), and then we apply the unsupervised method proposed by Conneau et al. (2017) to infer a bilingual dictionary which can be use for word-by-word translation.

Since WMT yields a very large-scale monolingual dataset, we obtain very high-quality embeddings and dictionaries, with an accuracy of $84.48\%$ and $77.29\%$ on French-English and German-English, which is on par with what could be obtained using a state-of-the-art supervised alignment method (Conneau et al., 2017).

On the Multi30k datasets instead, the monolingual training corpora are too small to train good word embeddings (more than two order of magnitude smaller than WMT). We therefore learn word vectors on Wikipedia using fastText[2].

---

[2]Word vectors downloaded from: `https://github.com/facebookresearch/fastText`

| | Multi30k-Task1 | | | | WMT | | | |
|---|---|---|---|---|---|---|---|---|
| | en-fr | fr-en | de-en | en-de | en-fr | fr-en | de-en | en-de |
| Supervised | 56.83 | 50.77 | 38.38 | 35.16 | 27.97 | 26.13 | 25.61 | 21.33 |
| word-by-word | 8.54 | 16.77 | 15.72 | 5.39 | 6.28 | 10.09 | 10.77 | 7.06 |
| word reordering | - | - | - | - | 6.68 | 11.69 | 10.84 | 6.70 |
| oracle word reordering | 11.62 | 24.88 | 18.27 | 6.79 | 10.12 | 20.64 | 19.42 | 11.57 |
| Our model: 1st iteration | 27.48 | 28.07 | 23.69 | 19.32 | 12.10 | 11.79 | 11.10 | 8.86 |
| Our model: 2nd iteration | 31.72 | 30.49 | 24.73 | 21.16 | 14.42 | 13.49 | 13.25 | 9.75 |
| Our model: 3rd iteration | 32.76 | 32.07 | 26.26 | 22.74 | 15.05 | 14.31 | 13.33 | 9.64 |

Table 2: **BLEU score on the Multi30k-Task1 and WMT datasets** using greedy decoding.

## 4.4 EXPERIMENTAL DETAILS

**Discriminator Architecture** The discriminator is a multilayer perceptron with three hidden layers of size 1024, Leaky-ReLU activation functions and an output logistic unit. Following Goodfellow (2016), we include a smoothing coefficient $s = 0.1$ in the discriminator predictions.

**Training Details** The encoder and the decoder are trained using Adam (Kingma & Ba, 2014), with a learning rate of 0.0003, $\beta_1 = 0.5$, and a mini-batch size of 32. The discriminator is trained using RMSProp (Tieleman & Hinton, 2012) with a learning rate of 0.0005. We evenly alternate between one encoder-decoder and one discriminator update. We set $\lambda_{auto} = \lambda_{cd} = \lambda_{adv} = 1$.

## 4.5 EXPERIMENTAL RESULTS

Table 2 shows the BLEU scores achieved by our model and the baselines we considered. First, we observe that word-by-word translation is surprisingly effective when translating into English, obtaining a BLEU score of 16.77 and 10.09 for *fr-en* on respectively Multi30k-Task1 and WMT datasets. Word-reordering only slightly improves upon word-by-word translation. Our model instead, clearly outperforms these baselines, even on the WMT dataset which has more diversity of topics and sentences with much more complicated structure. After just one iteration, we obtain a BLEU score of 27.48 and 12.10 for the *en-fr* task. Interestingly, we do even better than oracle reordering on some language pairs, suggesting that our model not only reorders but also correctly substitutes some words. After a few iterations, our model obtains BLEU of 32.76 and 15.05 on Multi30k-Task1 and WMT datasets for the English to French task, which is rather remarkable.

**Comparison with supervised approaches** Here, we assess how much labeled data are worth our two large monolingual corpora. On WMT, we trained the very same NMT architecture on both language pairs, but with supervision using various amounts of parallel data. Figure 4-right shows the resulting performance. Our unsupervised approach obtains the same performance than a supervised NMT model trained on about 100,000 parallel sentences, which is impressive. Of course, adding more parallel examples allows the supervised approach to outperform our method, but the good performance of our unsupervised method suggests that it could be very effective for low-resources languages where no parallel data are available. Moreover, these results open the door to the development of semi-supervised translation models, which will be the focus of future investigation. With a phrase-based machine translation system, we obtain 21.6 and 22.4 BLEU on WMT *en-fr* and *fr-en*, which is better than the supervised NMT baseline we report for that same amount of parallel sentences, which is 16.8 and 16.4 respectively. However, if we train the same supervised NMT model with BPE (Sennrich et al., 2015b), we obtain 22.6 BLEU for *en-fr*, suggesting that our results on unsupervised machine translation could also be improved by using BPE, as this removes unknown words (about 9% of the words in *de-en* are replaced by the unknown token otherwise).

**Iterative Learning** Figure 4-left illustrates the quality of the learned model after each iteration of the learning process in the language pairs of Multi30k-Task1 dataset, other results being provided in Table 2. One can see that the quality of the obtained model is high just after the first iteration

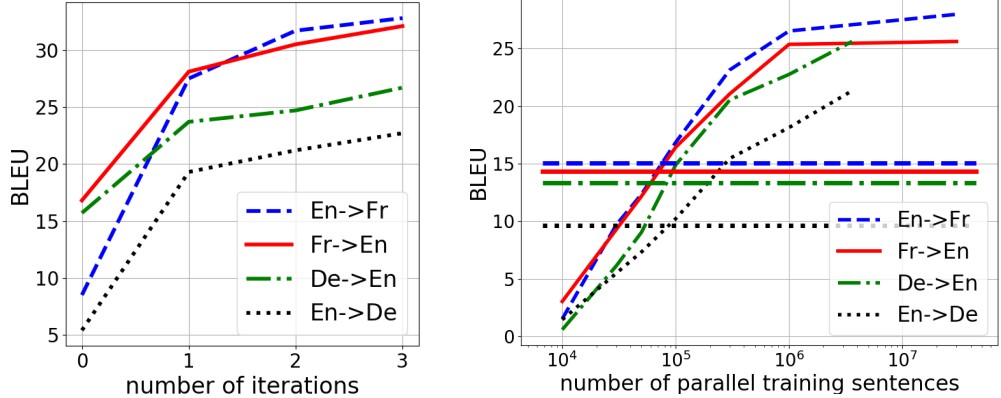

Figure 4: Left: BLEU as a function of the number of iterations of our algorithm on the Multi30k-Task1 datasets. Right: The curves show BLEU as a function of the amount of parallel data on WMT datasets. The unsupervised method which leverages about 15 million monolingual sentences in each language, achieves performance (see horizontal lines) close to what we would obtain by employing 100,000 parallel sentences.

| | |
|---|---|
| Source | un homme est debout près d' une série de jeux vidéo dans un bar . |
| Iteration 0 | a man is seated near a series of games video in a bar . |
| Iteration 1 | a man is standing near a closeup of other games in a bar . |
| Iteration 2 | a man is standing near a bunch of video video game in a bar . |
| Iteration 3 | a man is standing near a bunch of video games in a bar . |
| **Reference** | **a man is standing by a group of video games in a bar .** |
| Source | une femme aux cheveux roses habillée en noir parle à un homme . |
| Iteration 0 | a woman at hair roses dressed in black speaks to a man . |
| Iteration 1 | a woman at glasses dressed in black talking to a man . |
| Iteration 2 | a woman at pink hair dressed in black speaks to a man . |
| Iteration 3 | a woman with pink hair dressed in black is talking to a man . |
| **Reference** | **a woman with pink hair dressed in black talks to a man .** |
| Source | une photo d' une rue bondée en ville . |
| Iteration 0 | a photo a street crowded in city . |
| Iteration 1 | a picture of a street crowded in a city . |
| Iteration 2 | a picture of a crowded city street . |
| Iteration 3 | a picture of a crowded street in a city . |
| **Reference** | **a view of a crowded city street .** |

Table 3: **Unsupervised translations.** Examples of translations on the French-English pair of the Multi30k-Task1 dataset. Iteration 0 corresponds to word-by-word translation. After 3 iterations, the model generates very good translations.

of the process. Subsequent iterations yield significant gains although with diminishing returns. At iteration 3, the performance gains are marginal, showing that our approach quickly converges.

Table 3 shows examples of translations of three sentences on the Multi30k dataset, as we iterate. Iteration 0 corresponds to the word-by-word translation obtained with our cross-lingual dictionary, which clearly suffers from word order issues. We can observe that the quality of the translations increases at every iteration.

**Ablation Study** We perform an ablation study to understand the importance of the different components of our system. To this end, we have trained multiple versions of our model with some missing components: the discriminator, the cross-domain loss, the auto-encoding loss, etc. Table 4 shows that the best performance is obtained with the simultaneous use of all the described elements.

|  | en-fr | fr-en | de-en | en-de |
|---|---|---|---|---|
| $\lambda_{cd} = 0$ | 25.44 | 27.14 | 20.56 | 14.42 |
| Without pretraining | 25.29 | 26.10 | 21.44 | 17.23 |
| Without pretraining, $\lambda_{cd} = 0$ | 8.78 | 9.15 | 7.52 | 6.24 |
| Without noise, $C(x) = x$ | 16.76 | 16.85 | 16.85 | 14.61 |
| $\lambda_{auto} = 0$ | 24.32 | 20.02 | 19.10 | 14.74 |
| $\lambda_{adv} = 0$ | 24.12 | 22.74 | 19.87 | 15.13 |
| Full | **27.48** | **28.07** | **23.69** | **19.32** |

Table 4: **Ablation study on the Multi30k-Task1 dataset**.

The most critical component is the unsupervised word alignment technique, either in the form of a back-translation dataset generated using word-by-word translation, or in the form of pretrained embeddings which enable to map sentences of different languages in the same latent space.

On the English-French pair of Multi30k-Task1, with a back-translation dataset but without pretrained embeddings, our model obtains a BLEU score of 25.29 and 26.10, which is only a few points below the model using all components. Similarly, when the model uses pretrained embeddings but no back-translation dataset (when $\lambda_{cd} = 0$), it obtains 25.44 and 27.14. On the other hand, a model that does not use any of these components only reaches 8.78 and 9.15 BLEU.

The adversarial component also significantly improves the performance of our system, with a difference of up to 5.33 BLEU in the French-English pair of Multi30k-Task1. This confirms our intuition that, to really benefit from the cross-domain loss, one has to ensure that the distribution of latent sentence representations is similar across the two languages. Without the auto-encoding loss (when $\lambda_{auto} = 0$), the model only obtains 20.02, which is 8.05 BLEU points below the method using all components. Finally, performance is greatly degraded also when the corruption process of the input sentences is removed, as the model has much harder time learning useful regularities and merely learns to copy input data.

## 5 RELATED WORK

A similar work to ours is the style transfer method with non-parallel text by Shen et al. (2017). The authors consider a sequence-to-sequence model, where the latent state given to the decoder is also fed to a discriminator. The encoder is trained with the decoder to reconstruct the input, but also to fool the discriminator. The authors also found it beneficial to train two discriminators, one for the source and one for the target domain. Then, they trained the decoder so that the recurrent hidden states during the decoding process of a sentence in a particular domain are not distinguishable according to the respective discriminator. This algorithm, called Professor forcing, was initially introduced by Lamb et al. (2016) to encourage the dynamics of the decoder observed during inference to be similar to the ones observed at training time.

Similarly, Xie et al. (2017) also propose to use an adversarial training approach to learn representations invariant to specific attributes. In particular, they train an encoder to map the observed data to a latent feature space, and a model to make predictions based on the encoder output. To remove bias existing in the data from the latent codes, a discriminator is also trained on the encoder outputs to predict specific attributes, while the encoder is jointly trained to fool the discriminator. They show that the obtained invariant representations lead to better generalization on classification and generation tasks.

Before that, Hu et al. (2017) trained a variational autoencoder (Kingma & Welling, 2013) where the decoder input is the concatenation of an unstructured latent vector, and a structured code representing the attribute of the sentence to generate. A discriminator is trained on top of the decoder to classify the labels of generated sentences, while the decoder is trained to satisfy this discriminator. Because of the non-differentiability of the decoding process, at each step, their decoder takes as input the probability vector predicted at the previous step.

Perhaps, the most relevant prior work is by He et al. (2016), who essentially optimizes directly for the model selection metric we propose in section 3.2. One drawback of their approach, which has

not been applied to the fully unsupervised setting, is that it requires to back-propagate through the sequence of discrete predictions using reinforcement learning-based approaches which are notoriously inefficient. In this work, we instead propose to a) use a symmetric architecture, and b) *freeze* the translator from source to target when training the translator from target to source, and vice versa. By alternating this process we operate with a fully differentiable model and we efficiently converge.

In the vision domain, several studies tackle the unsupervised image translation problem, where the task consists in mapping two image domains A and B, without paired supervision. For instance, in the CoGAN architecture (Liu & Tuzel, 2016), two generators are trained to learn a common representation space between two domains, by sharing some of their convolutional layers. This is similar to our strategy of sharing the LSTM weights across the source and target encoders and decoders. Liu et al. (2017) propose a similar approach, based on variational autoencoders, and generative adversarial networks (Goodfellow et al., 2014). Taigman et al. (2016) use similar approaches for emoji generation, and apply a regularization term to the generator so that it behaves like an identity mapping when provided with input images from the target domain. Zhu et al. (2017) introduced a cycle consistency loss, to capture the intuition that if an image is mapped from A to B, then from B to A, then the resulting image should be identical to the input one.

Our approach is also reminiscent of the Fader Networks architecture (Lample et al., 2017), where a discriminator is used to remove the information related to specific attributes from the latent states of an autoencoder of images. The attribute values are then given as input to the decoder. The decoder is trained with real attributes, but at inference, it can be fed with any attribute values to generate variations of the input images. The model presented in this paper can be seen as an extension to the text domain of the Fader Networks, where the attribute is the language itself.

## 6 CONCLUSION

We presented a new approach to neural machine translation where a translation model is learned using monolingual datasets only, without any alignment between sentences or documents. The principle of our approach is to start from a simple unsupervised word-by-word translation model, and to iteratively improve this model based on a reconstruction loss, and using a discriminator to align latent distributions of both the source and the target languages. Our experiments demonstrate that our approach is able to learn effective translation models without any supervision of any sort.

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
