# OpenReview forum: "Unsupervised Machine Translation Using Monolingual Corpora Only"
_ICLR.cc/2018/Conference — Accept (Poster)_

### Official Review · AnonReviewer1 · 2017-11-27
**A thorough exploration of techniques for unsupervised translation, a very strong start for this problem**

**Rating:** 8
**Confidence:** 5

**Review:**

This paper describes an approach to train a neural machine translation system without parallel data. Starting from a word-to-word translation lexicon, which was also learned with unsupervised methods, this approach combines a denoising auto-encoder objective with a back-translation objective, both in two translation directions, with an adversarial objective that attempts to fool a discriminator that detects the source language of an encoded sentence. These five objectives together are sufficient to achieve impressive English <-> German and Engish <-> French results in Multi30k, a bilingual image caption scenario with short simple sentences, and to achieve a strong start for a standard WMT scenario.

This is very nice work, and I have very little to criticize. The approach is both technically interesting, and thorough in that it explores and combines a host of ideas that could work in this space (initial bilingual embeddings, back translation, auto-encoding, and adversarial techniques). And it is genuinely impressive to see all these pieces come together into something that translates substantially better than a word-to-word baseline. But the aspect I like most about this paper is the experimental analysis. Considering that this is a big, complicated system, it is crucial that the authors included both an ablation experiment to see which pieces were most important, and an experiment that indicates the amount of labeled data that would be required to achieve the same results with a supervised system.

In terms of specific criticisms:

In Equations (2), consider replacing C(y) with C(M(x)), or use compose notation, in order to make x-hat's relationship to x clear and self-contained within the equation.

I am glad you take the time to give your model selection criterion it's own section in 3.2, as it does seem to be an important part of this puzzle. However, it would be nice to provide actual correlation statistics rather than an anecdotal illustration of correlation.

In the first paragraph of Section 4.5, I disagree with the sentence, "Similar observations can be made for the other language pairs we considered." In fact, I would go so far as to say that the English to French scenario described in that paragraph is a notable outlier, in that it is the other language pair where you beat the oracle re-ordering baseline in both Multi30k and WMT.

When citing Shen et al., 2017, consider also mentioning the following:

Controllable Invariance through Adversarial Feature Learning; Qizhe Xie, Zihang Dai, Yulun Du, Eduard Hovy, Graham Neubig; NIPS 2017; https://arxiv.org/abs/1705.11122

Response read -- thanks.

---

> ### Author Response · Authors · 2017-12-29
> **response 1**
>
> We thank the reviewer for the feedback and comments.
>
> We clarified Equation (2), and also provided a correlation score between the unsupervised criterion and the actual test performance, thanks for the suggestion. The paper now contains: “The unsupervised model selection criterion is used both to a) determine when to stop training and b) to select the best hyper-parameter setting across different experiments. In the former case, the Spearman correlation coefficient between the proposed criterion and BLEU on the test set is 0.95 in average. In the latter case, the coefficient is in average 0.75, which is fine but not nearly as good. For instance, the BLEU score on the test set of models selected with the unsupervised criterion are sometimes up to 1 or 2 BLEU points below the score of models selected using a small validation set of 500 parallel sentences.”
>
> The work of Xie et al. was indeed relevant and we added it as a citation in the related work section of the updated version.
>
> As for the first paragraph of Section 4.5, we will clarify that “similar observations” refer to improvements as we iterate. Thank you for pointing this out.

---

### Official Review · AnonReviewer3 · 2017-11-28
**Need some clarifications**

**Rating:** 7
**Confidence:** 5

**Review:**

The authors present an approach for unsupervised MT which uses a weighted loss function containing 3 components: (i) self reconstruction (ii) cross reconstruction and (iii) adversarial loss. The results are interesting (but perhaps less interesting than what is hinted in the abstract).

1) In the abstract the authors mention that they achieve a BLEU score of 32.8 but omit the fact that this is only on Multi30K dataset and not on the more standard WMT datasets. At first glance, most people from the field would assume that this is on the WMT dataset. I request the authors to explicitly mention this in the abstract itself (there is clearly space and I don't see why this should be omitted)

2) In section 2.3, the authors talk about the Noise Model which is inspired by the standard Denoising Autoencoder setup.  While I understand the robustness argument in the case of AEs I am not convinced that the same applies to languages. Such random permutations can often completely alter the meaning of the sentence. The ablation test seems to suggest that this process helps. I read another paper which suggests that this noise does not help (which intuitively makes sense). I would like the authors to comment on this (of course, I am not asking you to compare with  the other paper but I am just saying that I have read contradicting observations - one which seems intuitive and the other does not).

3) How were the 3 lambdas in Equation 3 selected ? What ranges did you consider. The three loss terms seem to have very different ranges. How did you account for that?

4) Clarification: In section 2.5 what exactly do you mean by "as long as the two monolingual corpora exhibit strong structure in feature space." How do you quantify this ?

5) In section 4.1, can you please mention the exact number of sentences that you sampled from WMT'14. You mention that selected sentences from 15M random pairs but how many did you select ? The caption of one of the figure mentions that there were 10M sentences. Just want to confirm this.

6) The improvements are much better on the Multi30k dataset. I guess this is because this dataset has smaller sentences with smaller vocabulary. Can you provide a table comparing the average number of sentences and vocabulary size of the two datasets (Multi30k and WMT).

7) The ablation results are provided only for the Multi30k dataset. Can you provide similar results for the WMT dataset. Perhaps this would help in answering my query in point (2) above.

8) Can you also check the performance of a PBSMT system trained on 100K parallel sentences? Although NMT outperforms PBSMT when the data size is large, PBSMT might still be better suited for low resource settings.

9) There are some missing citations (already pointed by others in the forum) . Please add those.


+++++++++++++++++++++++
I have noted the clarifications posted by the authors. I still have concerns about a couple of things.  For example, I am still not convinced about the justification given for word order. I understand that empirically it works better but I don't get the intuition. Similarly, I don't get the argument about "strong structure in feature space". This is just a conjecture and it is very hard to measure it. I would request the authors to not emphasize on it or give a different more grounded intuition.

I do acknowledge the efforts put in by the authors to address some of my comments and for that I would like to change my rating a bit.

---

> ### Author Response · Authors · 2017-12-29
> **response 3**
>
> We thank the reviewer for the feedback and comments. We address each of them in turn:
>
> 1) It is true that the 32.8 BLEU score in the abstract was misleading, thank you for pointing this out. We updated the paper as follows: “We demonstrate our model on two widely used datasets and two language pairs, reporting BLEU scores of 32.8 and 15.1 on the Multi30k and WMT English-French datasets, without using even a single parallel sentence at training time.”
>
> 2) Modifying word orders and dropping words definitely alters the meaning of the sentence. However, without this component, we observed that the autoencoder was simply learning to copy the words in the input sentence one-by-one. Adding noise to the input sentence turned out to be an efficient solution that prevents the model from converging to that trivial solution. It is true that in some other tasks like sentence classification or machine translation, adding noise to the input sentence might alter the meaning of the sentence and deteriorate the overall performance of the system, but in the case of auto-encoding, this turned out to be necessary for us. In particular, on WMT en-fr / fr-en we obtain 5.24 / 6.57 without word shuffling (but with word dropout), and 1.69 / 5.54 when not using any form of noise. Please see response to 7) below for more results on the ablation study for WMT.
>
> 3) We also expected the tuning of these coefficients to be critical. We ran a few experiments with values from 1e-5 to 10, but in practice, we observed very small differences using different coefficients compared to fixing everything to 1.
>
> 4) “as long as the two monolingual corpora exhibit strong structure in feature space." This sentence was indeed incorrect, thank you for spotting this. We corrected it to “as long as the two latent representations exhibit strong structure in feature space”. What we meant is that if the learned word embeddings were iid distributed, then we would not be able to align them as any rotation would yield an equivalent matching of the two distributions. The reason why we can align well is because there are asymmetries which the algorithm exploits to align the two spaces. We are not aware of any study quantifying the structure in embedding space to the quality of the alignment. Here, we just meant to provide an intuition for our approach.
>
> 5) Thanks for noticing this, there was indeed a mistake in the caption. The unsupervised method uses 15M sentences (each for French and English, so 30M total) with WMT14 en-fr, and 3.6M sentences with WMT16 de-en (so 1.8M for each language).
>
> 6) We provided some statistics about the vocabulary size, and the number of monolingual sentences both for WMT and MMT1, in Table 1 of the updated version of the paper.
>
> 7) The experimental setup used in this paper is quite expensive, this is why we initially considered the much smaller MMT1 dataset to perform the ablations study, and to use that to decide on the best parameters for WMT. We ran new experiments to study the impact of each component when training on the WMT dataset, for the en-fr and fr-en language pairs. In particular, we obtain on en-fr / fr-en:
> - Without word shuffle, but with word dropout: 5.24 / 6.57
> - Without word shuffle, and without word dropout: 1.69 / 5.54
> - Without adversarial training: 10.45 / 10.29
> - Without auto-encoding: 1.20 / 1.21
> - Without word embeddings pretraining: 11.11 / 10.86
> - Without word embeddings, and without cross-domain training: 1.44 / 1.30
> - Without cross-domain training: 2.65 / 2.44
>
> 8) We ran experiments with a phrase-based system and obtained much better results than with a standard NMT system. Actually, our phrase-based models with 10k parallel sentences obtained 15.5 and 16.0 BLEU, which is roughly on par with what we report in the paper with our NMT model for 100k pairs. Note that our supervised NMT baseline in the paper is a bit weak, as we set a very large minimum word-count cutoff to reduce the vocabulary size and accelerate the experiments (and our unsupervised approach suffers from the same issue). We are currently running further experiments for other parallel corpora sizes and the de-en pair, and will report results with PBSMT in the paper very soon. Thank you for your suggesting this, these results are significantly better than what we expected, this will be very valuable to the paper, and opens new research directions.
>
> 9) We added the relevant citations as suggested in the comments.

---

### Official Review · AnonReviewer2 · 2017-11-30
**Remarkable paper -- limited modeling details**

**Rating:** 7
**Confidence:** 4

**Review:**

This paper introduces an architecture for training a MT model without any parallel material, and tests it on benchmark datasets (WMT and captions) for two language pairs. Although the resulting performance is only about half that of a more traditional model, the fact that this is possible at all is remarkable.

The method relies on fairly standard components which will be familiar to most readers: a denoising auto-encoder and an adversarial discriminator. Not much detail is given on the actual models used, for which the authors mainly refer to prior work. This is disappointing: the article would be more self-contained by providing even a high-level description of the models, such as provided (much too late) for the discriminator architecture.

Misc comments:

"domain" seems to be used interchangeably with "language". This is unfortunate as "domain" has another, specific meaning in NLP in general and SMT in partiular. Is this intentional (if so what is the intention?) or is this just a carry-over from other work in cross-domain learning?

Section 2.3: How do you sample permutations for the noise model, with the constraint on reordering range, in the general case of sentences of arbitrary lengths?

Section 2.5: "the previously introduced loss [...] mitigates this concern" -- How? Is there a reference backing this?

Figure 3: In the caption, what is meant by "(t) = 1"? Are these epochs only for the first iteration (from M(1) to M(2))?

Section 4.1: Care is taken to avoid sampling corresponding src and tgt sentences. However, was the parallel corpus checked for duplicates or near duplicates? If not, "aligned" segments may still be present. (Although it is clear that this information is not used in the algorithm)

This yields a natural question: Although the two monolingual sets extracted from the parallel data are not aligned, they are still very close. It would be interesting to check how the method behaves on really comparable corpora where its advantage would be much clearer.

Section 4.2 and Table 1: Is the supervised learning approach trained on the full parallel corpus? On a parallel corpus of similar size?

Section 4.3: What are the quoted accuracies (84.48% and 77.29%) measured on?

Section 4.5: Experimental results show a regular inprovement from iteration 1 to 2, and 2 to 3. Why not keep improving performance? Is the issue training time?

References: (He, 2016a/b) are duplicates

Response read -- thanks.

---

> ### Author Response · Authors · 2017-12-29
> **response 2**
>
> We thank the reviewer for the feedback and comments.
>
> We did not provide details about the architecture mainly because of lack of space, but we will add it in the updated version of the paper. Briefly, our architecture was composed of a standard encoder-decoder, with 3 LSTM layers, and an attention model without input-feeding. The embedding and LSTM hidden dimensions were set to 300.
>
> We now address the comments in turn:
>
> - We casted machine translation in the unsupervised setting as the problem to match distributions of latent features, which can be seen as a particular instance of domain adaptation where “domain” refers to a particular language. We will make sure to clarify this in the next version of the paper.
> - To generate random permutations with the reordering range constraint, we generate a random vector of the size of the sentence, and sort it by indexes. In NumPy, for a sentence of size n, it will look like: `  x = (np.arange(n) + alpha * np.random.rand(n)).argsort()'` Where alpha is a tunable parameter. alpha = 0 implies that x is the identity, alpha = infinity can return any permutation. With alpha = 4, for all i in [0, n[ we have |x[i] - i| <= 3. This has been added in the section 2.3 of the paper.
> - "the previously introduced loss [...] mitigates this concern": we are not aware of any reference about this, but this is the intuition we had while designing our loss function. The intuition is that auto-encoding with adversarial training ensures that latent representations of sentences in the two languages have similar distributions, but nothing constrains the system to actually translate (e.g., the sentence “je parle français” could be mapped into a latent space which could be decoded into the English sentence “the car is red”, which is a correct English sentence but not a good translation). However, the back-translation term does make sure that the latent representations actually (and eventually) correspond to each other, as the system has to produce a ground truth translation from a noisy source (and the auto-encoding term helps mapping noisy sentences into the same latent representation).
> - In the caption of Figure 3, “(t) = 1” indeed represents the training from M(1) to M(2). We clarified this in the updated version of the paper.
> - We did some experiments to investigate whether some duplicates of the removed sentences might be present among the selected sentences. To do so, we used a simple technique based on weighted bag-of-words embeddings, to retrieve the most similar sentences, and overall we have not been able to find very good matching pairs. We concluded that our two selected set of sentences were different enough in the sense that most sentences will not have an equivalent translation in the opposite language. However, it is true that the two domains remain similar. We are planning to investigate on the impact of the similarity of the two monolingual corpora on the translation quality in the future.
> - The supervised learning approach is trained on the full corpora (both for Multi30k and WMT).
> - The accuracies are measured on the word translation retrieval: given a test dictionary of 5000 pairs of words, we estimate how frequently a source word is properly mapped to its associated target word.
> - We ran a fourth iteration on Multi30k, but did not observe any improvement. The results would have improved by about 0.5/1 BLEU point if our unsupervised criterion had been perfect (see response to reviewer 1 about the quality of the criterion). However, using our criterion, this fourth iteration gave the same BLEU as the third iteration.

---

### Public Comment · (anonymous) · 2017-10-30
**More references in related work section**

The related work section can be improved by providing more references to earlier research on learning unsupervised alignment in different domains.

For example:
https://arxiv.org/abs/1611.02200 - Unsupervised cross domain image generation
https://arxiv.org/abs/1612.05424 - Unsupervised Pixel–Level Domain Adaptation with Generative Adversarial Networks
https://arxiv.org/abs/1606.03657 - InfoGAN: Interpretable Representation Learning by Information Maximizing Generative Adversarial Nets
https://arxiv.org/abs/1703.10593 - Unpaired Image-to-Image Translation using Cycle-Consistent Adversarial Networks

---

> ### Author Response · Authors · 2017-11-08
> **more references in related work section**
>
> Thank you for pointing out these references. We will definitely revise the paper accordingly. In particular, our model is reminiscent of CycleGAN, as well as other methods successfully applied in vision (such as the Coupled Generative Adversarial Networks of Liu et al.). The major conceptual difference is that we cannot easily chain as they do in these other works, because we deal with a discrete sequence of symbols as opposed to continuous vectors. Therefore, we resort to using the model at the previous iteration to produce translations in the other language, but we do not back-prop through this. The iterative nature of our approach together with the weight sharing between our encoder/decoders are the most important differences.

---

### Public Comment · (anonymous) · 2017-11-07
**Great work/related work**

This looks like great work, and I think merits a clear accept.

That said, I am also concerned about the lack of discussion of prior work. Almost all of the contributions which enabled the work are relegated to the companion article ("Word translation without parallel data"). While I understand the authors want to focus on their own contribution, the introduction/related work to a major achievement like this should convey how the community as a whole reached this point, especially since much of the progress was made outside of the big labs.

From following the citations, it seems like some key steps towards unsupervised translation were:

1. Monolingual high quality word vectors (https://arxiv.org/abs/1301.3781)
2. The linear transform for word translation from small dictionaries (https://arxiv.org/abs/1309.4168)
3. The orthogonal transform/SVD to improve resilience to low quality dictionaries (https://arxiv.org/abs/1702.03859, http://www.anthology.aclweb.org/D/D16/D16-1250.pdf)
4. The use of a GAN, regularized towards orthogonal transform, to obtain unsupervised bilingual word vectors (http://www.aclweb.org/anthology/P17-1179)
5. The iterative SVD procedure to enhance the GAN solution to supervised accuracy (http://www.aclweb.org/anthology/P17-1042)
6. The realization that aligned mean word vector provides a surprisingly good bilingual sentence space (https://arxiv.org/abs/1702.03859)
7. Finally the (significant) contribution of this work is to iterate this initial unsupervised shared sentence space towards a word order dependent translation model. A similar paper was submitted to ICLR simultaneously, "Unsupervised Neural Machine Translation".

My apologies to other important prior work I have missed!

---

> ### Author Response · Authors · 2017-11-30
> **related work**
>
> Thank you for your note. This paper indeed builds upon previous work, and we did our best to give credit to what we thought were the most relevant papers; in fact, we have 2 pages of citations already.
> As per your suggestion, we will add some of the references you mention. However, please keep in mind that our paper focuses on machine translation, while the references you pointed us at are more pertinent to the work on learning a bilingual dictionary.

---

### Public Comment · ~Meixue_Liu1 · 2017-12-15
**Reproducibility of the WBW baseline**

Our team reproduced word-by-word translation (WBW) baseline from the study. Based on our experiment, the WBW baseline is reproducible and we believe the whole study would be reproducible once the authors’ code is released to the public with further clarification on the BLEU score matric and hyperpaprameter section.

Dataset: Clear references to the datasets were provided, thus we were able to acquire them by a Google search. We used Multi30k-Task1 dataset for our experiment. Since the preprocessing steps were clearly explained, we obtained the same monolingual corpora for each language.

Code: All components of the code such as for fastText, BLEU score, and WBW were accessible. The authors provided the source of the code for fastText and a clear reference of the previous study on WBW. The code of WBW was published by the Facebook Research Team for the project MUSE which presents a word embedding model that can be trained either in a supervised or unsupervised way. The WBW code was implemented to datasets containing individual words. Because the dataset for the current study contain sentences, modification for the code was needed. To implement the code of WBW to Multi30k-Task1 dataset, we coded a method to translate each word of each sentence in the dataset. In addition, the BLEU score calculation package was found under the nltk.translate.api module.

Implementation: Since the size of the dataset is large and our personal computers were not able to efficiently perform the training, we used Google Cloud Platform to run the code on remote CPUs.
Our work focused on the unsupervised training, and the training process was smooth and successful. We were able to use Pytorch tensors to accelerate data processing on CPUs; We ran the code Python 2.7; We compiled Faiss with a Python interface and use it for our experiment.
The challenge is the settings for parameters and hyperparameters. The default settings of the hyperparameters come with the code of WBW for the study of Conneau et al. However, the tuned hyperparameters are not identified in the current study. We decided to focus on the reproducibility of methods and used the default settings in Conneau et al for our experiment.

Result: We were unable to obtain the same BLEU score as the study did. There might be two possible reasons. First, the hyperparameters used by the author or their implementation procedures are not the same as our experiment. It would be useful to present the values used for the baseline model. Second, according to Papineni et al. we learnt that the BLEU score metric normally ranges between 0 to 1 but the study presents scores that are not within the range. We suggest including an explanation on how the BLEU scores were calculated in order to improve the reproducibility.

Reference:
Papineni, Kishore, et al. "BLEU: a method for automatic evaluation of machine translation." Proceedings of the 40th annual meeting on association for computational linguistics. Association for Computational Linguistics, 2002.

---

> ### Author Response · Authors · 2018-01-03
> **wbw baseline**
>
> Hello, thank you for your note.
>
> We computed the BLEU score using the Moses perl script: https://github.com/moses-smt/mosesdecoder/blob/master/scripts/generic/multi-bleu.perl
> It is standard to multiply the score by 100 so that the BLEU is between 0 and 100 instead of 0 and 1, this is what the Moses script does and what we reported.
>
> Regarding the WBW baseline, it should be simple to reproduce because we used the default hyper-parameters in MUSE. Note that for the Multi30k dataset we used the fastText monolingual embeddings (we did not train them on the Multitask30k dataset because it’s too small).
> Finally, the translation is simply done word-by-word given the source reference file, and directly evaluated using the Moses script. Source words that are not in the dictionary were simply ignored.
>
> Please, let us know if you have other questions.
> Thanks.

---

### Comment · Area_Chair · 2017-12-27
**Discussion**

Authors,

Could I ask you to respond to the reviewers for discussion? While the reviewers here are quite positive, there are some points of clarification and concerns that would be nice to hash out.

---

### Author Response · Authors · 2018-01-05
**reviews**

We would like to thank the reviewers for all their comments and constructive feedback. We answered to each review individually, and uploaded a revision of the paper. Here is brief summary of the changes we made:
- updated some of the claims in the paper
- added details about the results in the abstract
- provided some statistics about the correlation between our unsupervised criterion and the BLEU test score
- added result with phrase based baseline
- added details about the model architecture
- explained how we generate random permutations with a constraint on the reordering
- simplified some notation
- clarified a few sentences / fixed typos
- added a table to compare the average number of sentences and the vocabulary size in the considered datasets
- added missing citations

---

### Decision · Program_Chairs · 2018-01-29
**ICLR 2018 Conference Acceptance Decision**

**Decision:**

Accept (Poster)

**Comment:**

This work presents some of the first results on unsupervised neural machine translation. The group of reviewers is highly knowledgeable in machine translation, and they were generally very impressed by the results and the think it warrants a whole new area of research noting "the fact that this is possible at all is remarkable.". There were some concerns with the clarity of the details presented and how it might be reproduced, but it seems like much of this was cleared up in the discussion. The reviewers generally praise the thoroughness of the method, the experimental clarity, and use of ablations. One reviewer was less impressed, and felt more comparison should be done.